# ALPHAOPT: FORMULATING OPTIMIZATION PROGRAMS WITH SELF-IMPROVING LLM EXPERIENCE LIBRARY

## ABSTRACT

Optimization modeling enables critical decisions across industries but remains hard to automate: informal language must be mapped to precise mathematical formulations and executable solver code, while prior LLM approaches either rely on brittle prompting or costly retraining with limited generalization. We present **AlphaOPT**, a self-improving *experience library* that enables an LLM to learn from limited demonstrations (i.e, even answers along without gold-standard program) and solver feedback without annotated reasoning traces or parameter updates. AlphaOPT operates a continual two-phase cycle: (i) a *Library Learning* phase that reflects on failed attempts, extracts solver-verified, structured insights as {*taxonomy*, *condition*, *explanation*, *example*}; and (ii) a *Library Evolution* phase that diagnoses retrieval misalignments and refines the applicability conditions of stored insights, improving transfer across tasks. This design (1) learns efficiently from limited demonstrations without curated rationales, (2) expands continually without costly retraining by updating the library rather than model weights, and (3) makes knowledge explicit and interpretable for human inspection and intervention. Experiments show that AlphaOPT steadily improves with more data (65% → 72% from 100 to 300 training items) and surpasses the strongest baseline by 7.7% on the out-of-distribution OptiBench dataset when trained only on answers.

## 1 INTRODUCTION

Optimization models support critical decision-making in finance, manufacturing, marketing, transportation, and logistics (AhmadiTeshnizi et al., 2023; Bertsimas & Tsitsiklis, 1997; Ramamonjison et al., 2022). Beyond improving efficiency, automating the optimization workflow lowers the barrier to operations research expertise in industry, enabling non-experts to prototype faster, iterate on formulations, and deploy solver-backed decisions at scale. Yet this process has long been challenging, as informal and often ambiguous specifications must be mapped to precise, domain-specific formulations and paired with appropriate code and solvers, creating major bottlenecks for end-to-end automation (Jiang et al., 2025).

Advances in large language models (LLMs) make this vision increasingly feasible: they can parse natural language requirements (Ouyang et al., 2022), generate executable programs (Nijkamp et al.; Jimenez et al., 2024), and orchestrate downstream tools (Qin et al., 2024). Two main lines of work have emerged. Prompt-based systems steer general LLMs with structured prompts and tool use (Xiao et al., 2023; Thind et al., 2025; AhmadiTeshnizi et al., 2024; Zhang & Luo, 2025). Fine-tuning approaches adapt models on domain corpora and benchmarks (Huang et al., 2025; Yang et al., 2024). Despite this progress, both families face limitations: prompt-based systems stop improving once they run out of fixed templates, and they are fragile to small wording changes and shifts in the domain; fine-tuned models require costly retraining and, critically, most benchmarks and datasets in the community (e.g., NLP4LP (AhmadiTeshnizi et al., 2024), MAMO (Huang et al., 2024), IndustryOR (Huang et al., 2025)) contain only programs/solutions rather than the intermediate reasoning that governs modeling choices, thereby limiting the generalizability of fine-tuning approaches. This motivates a new learning paradigm for optimization formulation: instead of rely-

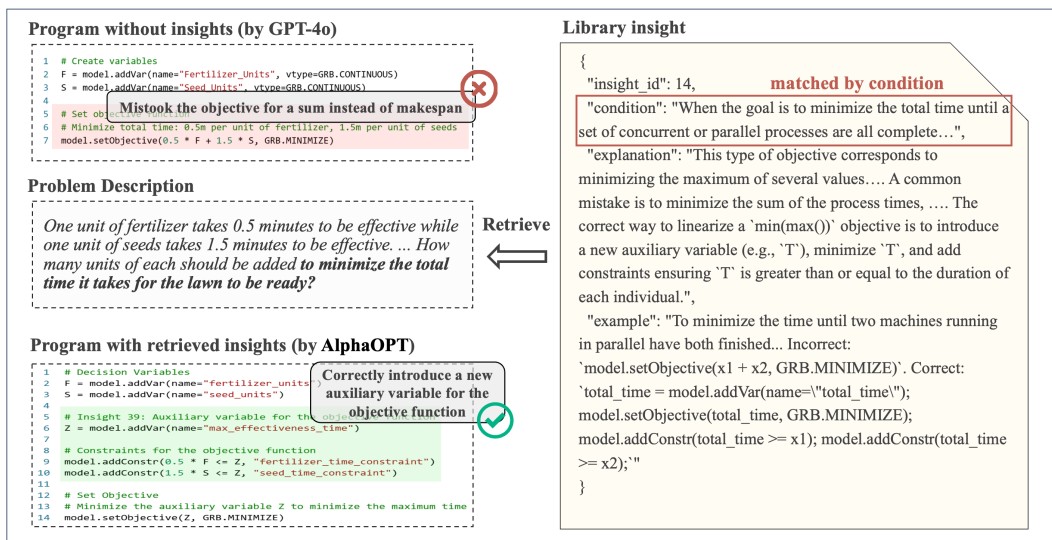

Figure 1: AlphaOPT identifies and retrieves relevant insights to guide problem solving. In this example, it avoids the common mistake of minimizing the sum of process times and instead introduces an auxiliary variable to correctly minimize the makespan, leading to the correct solution.

ing solely on prompts or retraining, LLMs should continually improve by accumulating, refining, and reusing solver-verified modeling insights.

We propose **AlphaOPT**, a self-improving framework that builds and refines a structured library of solver-verified insights for optimization formulation, as exemplified in Figure 1. Each insight encodes a reusable modeling rule in the form of a 4-tuple (*taxonomy*, *(applicability) condition*, *explanation*, *example*), which specifies not only what to reuse but also when and why it applies. We remark that our library learning framework does not require backpropagation to update framework parameters and can be regarded as the evolutionary mechanism. **AlphaOPT** improves through a continual two-phase cycle. Library Learning acquires new insights from both gold programs (when available) and solver-verified answer-only supervision, organizing them into a dynamically updating hierarchical taxonomy. Library Evolution then diagnoses misalignments between tasks and insight applicability, and refines conditions using aggregate evidence, ensuring that insights remain neither too narrow nor overly general. This design yields a principled optimization view: library construction corresponds to maximizing expected task success induced by task–insight matching while regularizing size to maintain efficiency and prevent redundancy.

We conduct quantitative experiments across multiple benchmarks and baselines, as well as qualitative analyses of the learned library. The results show that, compared to prompt-based or fine-tuning approaches, **AlphaOPT** (1) learns efficiently from limited demonstrations (i.e., it can learn from answers without recalling formulation) without requiring annotated reasoning traces or even gold-standard programs, (2) achieves stronger out-of-distribution generalizability and more consistent continual growth than learning-based methods, and (3) makes knowledge explicit and interpretable for human inspection and involvement.

Beyond these advantages, **AlphaOPT** also achieves state-of-the-art performance on multiple benchmarks. These results demonstrate the efficacy and potential of self-improving experience-library learning for optimization formulation, paving the way toward more challenging settings, such as efficient program formulation and large-scale optimization.

Our main contributions are threefold:

- **A library learning framework that learns from answers only.** We propose the first experience-library learning framework for natural language optimization formulation tasks, formally grounded in a mathematical view. The system can learn solely from answers, without requiring gold-standard programs.

- **A reusable and interpretable experience library.** We construct the first solver-verified library of structured modeling insights for LLM systems, designed to be reusable across tasks and explicitly interpretable for reliable transfer in operations research domains.

- **State-of-the-art out-of-distribution generalization.** AlphaOPT achieves strong generalization beyond training distributions, attaining state-of-the-art performance on LogiOR and OptiBench benchmarks.

## 2 RELATED WORK

**LLMs for Solving Optimization Problems.** Related work can be categorized into prompt-based and learning-based approaches. Prompt-based methods guide reasoning and modeling through multi-step prompts using proprietary LLMs (AhmadiTeshnizi et al., 2024; Xiao et al., 2023). AhmadiTeshnizi et al. (2023) first introduced OptiMUS, demonstrating how LLMs can generate optimization formulations from natural language descriptions, and OptiMUS-0.3 (AhmadiTeshnizi et al., 2024) extends this line of work to large-scale instances, introducing retrieval-augmented prompting and solver-integrated verification to improve scalability.

In contrast, learning-based methods construct training datasets and apply instruction tuning to open-source LLMs. Huang et al. (2025) proposed a semi-automated pipeline to synthesize training data, which is then used to fine-tune an open-source ORLM model. LLMOPT (Jiang et al., 2025) combines both paradigms by modeling optimization with five elements and fine-tuning on expert-annotated data via multi-instruction SFT. More recently, ORThought (Yang et al., 2025a) introduced the LogiOR benchmark and an expert-guided chain-of-thought framework, providing a systematic dataset and evaluation pipeline for optimization tasks that require harder logic. In terms of multi-agent design, Xiao et al. (2023) employs a collaborative multi-expert framework to enhance reasoning, Zhang & Luo (2025) integrates sandbox-based code execution and self-repair/self-verification.

Several benchmarks now exist that cover LP, MILP, NLP, and other optimization problem types (Xiao et al., 2023; AhmadiTeshnizi et al., 2024; Huang et al., 2025; Yang et al., 2024). Yet, none of the prior work has investigated strengthening LLMs' optimization capabilities by *learning and reusing structured modeling experience*.

**Decision-making tasks with Library Learning.** Library Learning refers to the process where reusable patterns or modules are automatically extracted from past experiences to improve future problem-solving. These experiences include concrete trajectories or demonstrations, as well as abstracted rules generalized from successful or failed attempts (Zhao et al., 2024; Mu et al., 2025; Feng et al., 2025; Wang et al.; Zhu et al., 2023). In terms of experience improvement, Zhao et al. (2024) and Mu et al. (2025) leverage an LLM to prune the library by checking if a newly added insight duplicates or conflicts with existing insights, or merges and generalizes from those overlapping insights. Feng et al. (2025) uses check functions to ensure that LLM-translated action sequences remain within the generalization boundaries of the original experiences.

**LLM-driven Evolutionary Methods.** Recent LLM-driven evolutionary frameworks have achieved remarkable advances in scientific discovery, showcasing LLM's capacity for broad generative exploration on solutions. Romera-Paredes et al. (2024) introduces FunSearch, a genetic programming driven by LLMs to search for feasible or optimal solutions of mathematical problems. AlphaEvolve (Novikov et al., 2025) extends the FunSearch system to provide the ability to perform multiobjective optimization using rich forms of natural-language context and feedback. Grayeli et al. (2024) applies LLMs to discover abstract concepts from high-performing hypotheses, combining symbolic regression with LLM-guided exploration within a concept library. ReEvo (Ye et al., 2024) frames LLMs as hyper-heuristics with a reflective evolution mechanism, enabling the generation and iterative refinement of heuristics across multiple NP-hard problems. HeurAgenix (Yang et al., 2025b) further develops this direction by evolving a pool of heuristics and dynamically selecting the most suitable one for each problem state, highlighting LLMs' role in adaptive heuristic discovery. Besides, LLM-guided evolution has also found use in discovering heuristics for combinatorial optimization (Liu et al.).

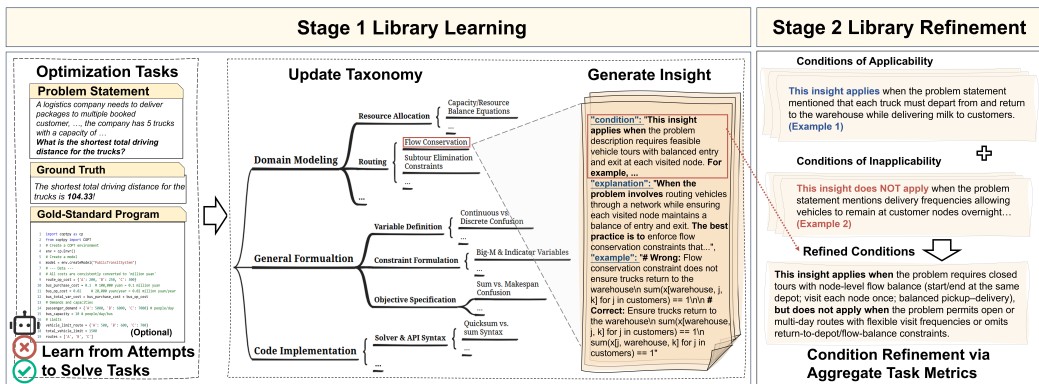

Figure 2: AlphaOPT builds and refines an experience library over multiple iterations. Each iteration consists of a library learning stage, which produces insights from attempts to solve tasks, and a library refinement stage, which adjusts the applicability of insights to avoid being too narrow or too general.

# 3 METHODOLOGY

Optimization tasks arrive with diverse natural-language descriptions, yet they share recurring modeling rules that activate under identifiable conditions. We identify three major challenges in building reliable systems that leverage LLMs to formulate and solve optimization problems using existing technologies and resources. First, gold-standard programs are scarce and may contain annotation errors (Jiang et al., 2025; Yang et al., 2025a), while datasets with only answer labels remain underutilized (Huang et al., 2024; 2025; Lu et al., 2025). Second, fine-tuned models (Huang et al., 2025; Jiang et al., 2025) struggle to generalize because the crucial when-to-apply-what knowledge is weakly represented (or missing) in training data; they can mimic syntax without mastering applicability. Third, the performance of prompt-based agent systems AhmadiTeshnizi et al. (2023); Xiao et al. (2023); Yang et al. (2025a) stagnates as the number of exemplars increases: they rely on human empirical curation and lack the capacity to adapt or to continually learn from larger datasets.

We propose AlphaOPT, an experience-library learning framework that learns from both gold programs (when available) and answer-only supervision. AlphaOPT iteratively builds a structured, solver-verified repository of reusable insights with explicit applicability conditions and evolves these conditions at the population level to improve generalization while avoiding redundancy. This two-stage design is described in Section F.2. In Section 3.2, we provide a mathematical interpretation that frames library construction as maximizing task success with a size regularizer. In Section A, we compare our method with prior works on learning from experience and self-evolving problem-solving agents.

## 3.1 ALPHAOPT FRAMEWORK

The framework incrementally learns a structured library of experiences over iterations until a stopping criterion indicates that the current model can no longer make meaningful improvements. As illustrated in Figure 2, each iteration consists of two complementary phases that form a continual cycle of acquisition and refinement. The first phase, Library Learning, extracts insights from individual tasks under either gold-program or answer-only supervision while minimizing redundancy. The second phase, Library Evolution, diagnoses misalignments between insights and tasks and refines applicability conditions to enhance generalization while reducing confusion caused by overgeneralization. The design follows three guiding principles: it is failure-driven (every error becomes a learning opportunity), locally validated (an insight must solve its source task before being admitted), and compact yet generalizable (redundant insights are merged and conditions refined to prevent uncontrolled growth that hinders retrieval and execution). The prompts for all LLM modules are provided in Appendix F.4.

### 3.1.1 Library Learning

The objective of this stage is to generate reusable insights as structured 4-tuples (*Taxonomy*, *Condition*, *Explanation*, *Example*) and organize them in a hierarchical taxonomy for efficient retrieval, while minimizing redundancy in the library. The workflow for this stage is illustrated in Figure 6.

**Insight Extraction, Representation, and Supervision Mode.** Insights can be learned from either problems with a gold-standard program or with the answer alone. For each task, the system first constructs a mathematical formulation, then generates an executable solver program, and invokes the solver. When the library is non-empty, both steps are guided by retrieved insights. If the generated program does not achieve the correct optimal value, two supervision modes are used to guide the generation of new insights. When a *gold program* is available, the system compares the candidate formulation and program against the reference, diagnosing discrepancies (e.g., missing variables, misformulated constraints, incorrect objective terms) and distilling them into insights. When only the *answer* (i.e., final optimal objective) is provided, the system performs solver-guided self-exploration: it iteratively proposes executable programs, reuses prior failures as context, and receives verification from the solver. Once a program achieves its correct objective, it is treated as a proxy for the gold standard in anchor insight extraction. Before being stored in the library, each insight is locally verified by reapplying it to its source task to ensure that it resolves the original failure. In addition, to mitigate stochastic successes that could obscure useful lessons, we conduct three independent trials per task, allowing errors from probabilistic generation to serve as signals for learning.

Each insight is represented as a structured 4-tuple: *Taxonomy*, hierarchical labels for indexing and retrieval; *Condition*, an explicit description of the applicability signals in the problem; *Explanation*, the underlying principle of applying this insight; and *Example*, a concrete demonstration such as a mathematical constraint or code snippet.

**Library Storage and Retrieval.** Insights are stored in a dynamically updated hierarchical taxonomy organized into three main tracks: *Domain Modeling* (problem-specific structures and assumptions), *General Formulation* (reusable mathematical patterns), and *Code Implementation* (solver-specific coding practices). Under each track, insights are further classified with two-level labels, where Level-1 captures a broad category and Level-2 refines it into a more specific subcategory. The taxonomy is initialized with few-shot labels and expands online: each new insight is either mapped to an existing category or, if no suitable label exists, prompts the LLM to propose new Level-1 or Level-2 labels. Each label is also assigned a condition, written by the LLM, that specifies when the category should be retrieved. When storing insights, to reduce redundancy, the LLM also checks whether a similar insight already exists and performs merging when appropriate. To align a target task with relevant insights, we employ a two-step LLM-driven retrieval procedure: Quick label matching, then full applicability check. The system first scans the taxonomy dictionary to identify labels that are potentially relevant to the context of the tasks. For example, Level-2 label such as Fixed Charge (Big-M Linking) will be probably detected when the problem description specifies that service or flow is allowed only if a facility is opened. After candidate labels are identified, the system rigorously evaluates each associated insight by examining its condition, and only the most applicable insights are retained.

During solution generation, retrieved insights from the *Domain Modeling* and *General Formulation* tracks guide the construction of the mathematical model, while insights from the *Code Implementation* track guide solver-code generation. This two-step procedure ensures that insights are applied appropriately, while the hierarchical taxonomy provides an extensible structure for organizing and retrieving insights as the library grows.

**Operational Flow.** Training proceeds in an online regime over minibatches of data, starting from an empty library. For each batch, the system retrieves candidate insights, generates and executes programs, and upon failures extracts insights and immediately commits those that pass local self-verification, allowing newly added insights to benefit subsequent tasks and preventing the generation of repetitive insights. To reduce redundancy, tasks are clustered and reordered by problem type and semantic similarity, and overlapping insights within a batch are merged prior to integration. The process iterates until overall accuracy plateaus, at which point the library is archived and used for evaluation.

In implementation, for the sake of efficiency, training follows two coordinated data flows. The first processes minibatches of tasks in parallel for insight extraction. The second maintains a centralized queue of all generated insights, storing them into the library sequentially. This queue does not allow asynchronous updates, as concurrent modifications could lead to conflicts if two insights attempt to update the library simultaneously. This design balances parallelism in problem-solving with strict serialization in library updates, ensuring both efficiency and consistency.

### 3.1.2 LIBRARY EVOLUTION

While Library Learning expands the repository of insights, Library Evolution aims to transform task-specific lessons into broadly applicable knowledge. Since each insight's applicability is defined by a condition induced from a specific task, early conditions are often too narrow (failing to trigger on relevant tasks) or too broad (causing misretrieval). Left unchecked, these misalignments lead to missed opportunities or systematic failures. Library Evolution counters this with a diagnostic–refinement cycle: it detects misaligned insights, aggregates evidence across tasks, and refines conditions at the end of each iteration. The refinement is guided by an aggregate metric rather than ad-hox fixes. As illustrated in Figure 3, library refinement can be understood as adjusting each insight's condition toward the correct retrieval boundary in the problem space. The workflow for this stage is illustrated in Figure 7.

**Diagnostic: Library Diagnosis.** After each training round, we trace failed tasks and analyze their interaction with the library. The diagnostic agent partitions the relationship between each insight $i$ and its associated tasks into three disjoint categories: $\Pi(i) = \{\text{Positive} : S_i^+, \text{Negative} : S_i^-, \text{Unretrieved} : S_i^u\}$ where $S_i^+$ contains tasks where the insight was applicable and contributed to the correct formulation, $S_i^-$ contains tasks where it was misleading and degraded performance, and $S_i^u$ contains tasks where it was not retrieved but would have been beneficial. By maintaining these partitions across iterations, the system continuously builds a performance profile for each insight. If a failed task is subsequently solved after removing a misleading (negative) insight or by injecting a previously unretrieved one, the system attributes the failure to condition misalignment rather than lack of knowledge, thus avoiding redundant insight generation. Unretrieved tasks are identified by first

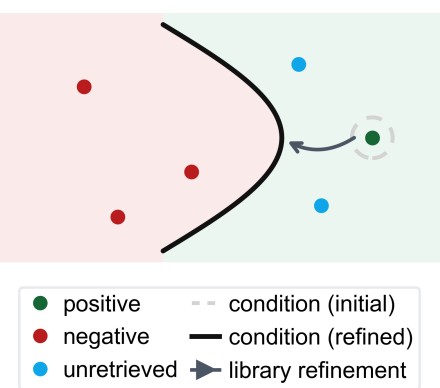

positive — condition (initial)
negative — condition (refined)
unretrieved — library refinement

Figure 3: A locally verified *initial* condition (light-grey dashed circle) is refined into a broader *applicability* boundary (solid black) through evolutionary prompt optimization guided by the aggregate metric.

comparing the model's generated program with the ground-truth (or a reference program from self-exploration) to locate discrepancies. These discrepancies guide the search for candidate insights, which are then verified for their ability to resolve the errors. Verified insights are flagged as relevant but unretrieved, allowing the system to diagnose retrieval gaps without resorting to intractable combinatorial search.

**Evolver: Library Refinement.** Building on the diagnosis, the Evolver agent refines insights in two steps: condition refinement and refinement verification. First, for each diagnosed insight, the agent strengthens or prunes its applicability condition. Negative tasks contribute explicit *inapplicability clauses* (e.g., constraints or contexts that block use), while unretrieved tasks highlight missing applicability signals. The Evolver then proposes multiple refinement strategies (e.g., adding preconditions, introducing keyword anchors, merging overlapping triggers) and produces candidate conditions with the goal of preserving correct cases, eliminating mismatches, and recovering previously missed tasks. Then, each candidate condition replaces the original and is tested over the union $R_i = S_i^+ \cup S_i^- \cup S_i^u$. A performance score

$$p_i = \frac{|\text{kept positives}| + |\text{corrected negatives}| + |\text{recovered unretrieved}|}{|R_i|}$$

quantifies improvement. Here, "kept positives" are tasks that still remain correctly retrieved after refinement; "corrected negatives" are tasks that were misled by the insight before and no longer retrieved; and "recovered unretrieved" are tasks that become correctly retrieved after refinement. We accept refinements that increase $p_i$ and keep the one with the highest $p_i$.

## 3.2 OPTIMIZATION PERSPECTIVE

The framework can be viewed as an iterative solution to the optimization problem in the library space. Let $\mathcal{L}$ denote a candidate library and $\mathcal{T}$ the distribution of the optimization problems we want to solve. The objective is to maximize task success while penalizing library complexity to mitigate retrieval inefficiency and long-context degradation in LLM inference:

$$\max_{\ell \in \mathcal{L}} \ \mathbb{E}_{t \sim \mathcal{T}}[\text{Success}(t \mid \ell)] \ - \ \lambda \, \Omega(\ell).$$

where $\text{Success}(t \mid \ell)$ indicates whether $\ell$ enables the system to produce a program that achieves the correct optimal objective for task $t$, and $\Omega(\ell)$ quantifies library complexity (e.g., number of insights or redundancy-adjusted size). Under our problem design—bounded and continuous property of $\text{Success}(\cdot)$ and $\Omega(\cdot)$, sufficient exploration under solver verification, and bounded merging—the refinement dynamics converge to a locally optimal library. In Appendix D, we provide a conceptual sketch showing that convergence holds: As refinement in the second phase strictly improves the first term, while verified merging in the first phase reduces the second term without diminishing the first, sufficient exploration combined with iterative cycles of library learning and evolution ensures convergence to a local optimum. Given the inherent ambiguity of natural language and stochasticity in LLM outputs, we present this perspective not as a strict theorem but as a principled justification for the acquisition–refinement design and the redundancy-reduction operations.

## 4 EXPERIMENTS

Our experiments are designed to reflect the requirements that arise in real-world optimization and operations research (OR) applications. In these settings, methods are expected not only to perform well on standard benchmarks, but also to transfer across domains, to remain effective when limited supervision is available, to improve steadily as more data becomes available, and to offer results that can be inspected and audited. We therefore organize our evaluation around four questions: (1) How well does the method generalize across domains? (2) Can it learn effectively with limited supervision? (3) Does performance improve consistently with more training data? (4) How does it compare overall with strong baselines? Finally, we examine the interpretability of the insight library to assess whether the outputs are understandable and actionable to practitioners.

### 4.1 EXPERIMENTAL SETUP

Our experiments are conducted on a dataset of 454 problem instances, aggregated from four real-world optimization and operation task datasets, namely the NLP4LP (AhmadiTeshnizi et al., 2024), NL4OPT (Ramamonjison et al., 2022), IndustryOR (Huang et al., 2025), MAMO (ComplexLP) (Huang et al., 2024), with any invalid entries discarded. These collections span various formulation types and originate from diverse sources, including academic papers, textbooks, and real-world industry scenarios. Detailed descriptions of these datasets are provided in Appendix B.

We perform stratified sampling within each dataset, randomly partitioning 70% for training and 30% for testing. We maintain a strict separation between training and test data. The experience library is constructed only from training tasks, ensuring that training-derived insight examples do not leak into the test set. To assess out-of-distribution generalization, we additionally evaluate on LogiOR (Yang et al., 2025a) and Optibench (Yang et al., 2024).

**Baselines.** We evaluate against two families of baselines. (i) **Prompt-based**: a vanilla baseline that directly generates the mathematical model from a simple prompt, as well as Reflexion (Shinn et al., 2023), OptiMUS (AhmadiTeshnizi, Gao, and Udell, 2024), and ORThought (Yang et al., 2025a). (ii) **Learning-based**: ORLM (Huang et al., 2025), built on LLaMa3-8B, and LLMOPT (Jiang et al., 2025), built on Qwen2.5-14B (the latest open-source version available after their paper).

## 4.2 Out-of-Distribution Generalization

We evaluate how well different methods generalize beyond their training distribution. For this purpose, we use two benchmarks that were not included during training: LogiOR (Yang et al., 2025a) and OptiBench (Yang et al., 2024).

These datasets were either released after the baseline model (ORLM) or explicitly designed in baseline model's experiment setting to avoid overlap with their training set (LL-MOPT).

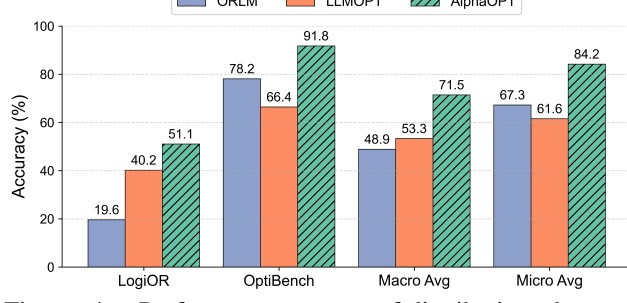

Figure 4: Performance on out-of-distribution datasets. Numbers report test accuracy on LogiOR, OptiBench, and OptMath.

Figure 4 summarizes the results. Fine-tuned models such as ORLM and LLMOPT show strong in-distribution performance but exhibit a noticeable drop on unseen datasets. For example, ORLM falls to 19.6% on LogiOR and 13.3% on OptMath, while LLMOPT performs better but still degrades compared to its in-distribution performance. By contrast, AlphaOPT maintains higher accuracy across all three benchmarks, reaching 51.1% on LogiOR and 91.8% on OptiBench. These results support our hypothesis: fine-tuned models tend to learn the syntax of solutions and may perform well when problems are very similar, but they struggle to capture the underlying principles needed for broader problem solving. In contrast, the learned experience library equips **AlphaOPT** with stronger out-of-distribution generalization capability.

## 4.3 Learning with Limited Supervision

In practical OR applications, gold-standard programs are rarely available. We therefore test whether AlphaOPT can learn solely from answers. Since two datasets in our training set contain gold-standard programs, we remove them in this experiment and let **AlphaOPT** learn exclusively from answer labels through self-exploration, as introduced in Section 3.1.1. As shown in the last two rows of Table 2, remarkably, when trained with answer-only supervision, **AlphaOPT** achieves accuracy comparable to when it is trained with gold-standard programs. **AlphaOPT** (self-explore) outperforms all prompt-based methods on test splits of the training data and even achieves the best performance on the OOD OptiBench dataset (92.1% accuracy). This demonstrates another advantage of **AlphaOPT** over fine-tuning–based methods, which require detailed annotations of mathematical formulations and code in order to achieve strong performance.

## 4.4 Continual Growth with Data

We test whether **AlphaOPT** can improve its performance as more data becomes available. We incrementally sample sets of 100, 200, and 300 data items from our training set and train **AlphaOPT** on each subset. As shown in Table 1, when evaluated on out-of-distribution datasets (LogiOR, OptiBench), we observe that **AlphaOPT** steadily improves its performance with increasing data size, without requiring updates to its model parameters.

Table 1: AlphaOPT steadily improves in both Micro and Macro averages with increasing training size, while maintaining a compact library.

| Training Size | MicroAvg | MacroAvg | Library Size |
|---|---|---|---|
| 100 | 83.24% | 65.80% | 38 |
| 200 | 85.09% | 69.22% | 103 |
| 300 | 85.21% | 72.12% | 110 |

## 4.5 Overall Performance

**AlphaOPT** achieves the best accuracy on out-of-distribution datasets, reaching 51.1% on LogiOR and 91.8% on OptiBench. On in-distribution test splits, fine-tuned models such as ORLM and LLMOPT achieve higher scores on certain datasets (e.g., LLMOPT obtains 97.3% on NLP4LP and

Table 2: Accuracy on in-distribution *Test Split* and *Out-of-Distribution* datasets (higher is better). Best per column in **bold**.

| Method | | Test Split | | | | Out-of-Distribution | |
|---|---|---|---|---|---|---|---|
| | | NLP4LP (73) | NL4OPT (64) | IndustryOR (25) | MAMO (ComplexLP) (34) | LogiOR (92) | OptiBench (403) |
| *Prompt-based* | Standard | 68.5 | 54.7 | 52.0 | 44.1 | 46.7 | 72.7 |
| | Reflexion | 76.7 | 64.1 | 56.0 | 47.1 | 43.5 | 76.9 |
| | OptiMus | 71.2 | 73.4 | 36.0 | 29.4 | 17.4 | 74.7 |
| | ORThought | 69.9 | 75.0 | **60.0** | 41.2 | 44.6 | 84.4 |
| *Fine-tuning-based* | ORLM | 86.3 | **87.5** | 36.0 | 55.9 | 19.6 | 78.2 |
| | LLMOPT | **97.3** | 86.5 | 44.0 | **85.8** | 40.2 | 66.4 |
| *Ours* | AlphaOPT (full) | 83.6 | 79.7 | **60.0** | 85.3 | **51.1** | 91.8 |
| | AlphaOPT (self-explore) | 86.3 | 79.7 | **60.0** | 76.5 | 50.0 | **92.1** |

85.8% on MAMO). However, these advantages are less conclusive, since LLMOPT's training data are not publicly available and may overlap with our test splits. Moreover, many existing benchmarks are derived from a small set of seed problems (Ramamonjison et al., 2022; Huang et al., 2024), which favors fine-tuning approaches that excel at pattern memorization. In contrast, **AlphaOPT** performs competitively across all in-distribution datasets, matches or exceeds baselines on IndustryOR and MAMO(ComplexLP), and maintains a clear margin on out-of-distribution generalization. These results demonstrate that the experience library enables **AlphaOPT** to learn transferable modeling principles rather than dataset-specific syntax, resulting in stronger robustness to distribution shifts.

### 4.6 LIBRARY ANALYSIS

To fully interpret the content and structure of the library learned from the training data, we visualize the distribution of insights by taxonomy across the three tracks (Figures 5) and discuss the library distribution in Section E.

## 5 CONCLUSION AND DISCUSSION

This paper addresses the limitations of previous methods by presenting a novel self-improving library learning framework, **AlphaOPT**, for formulating optimization programs. **AlphaOPT** can learn from answer labels only, achieves much stronger out-of-distribution generalization than fine-tuning–based methods, and provides interpretable and auditable structured knowledge to support human involvement in real-world practice.

Looking ahead, we highlight three promising directions. First, reasoning-oriented test-time scaling, which is already powerful in other domains, could be particularly effective for OR formulations, where results are inherently verifiable. Second, strengthening datasets with both academic research and large-scale real-world industry problems will move LLM systems beyond the toy examples that dominate current benchmarks, enabling progress toward truly large-scale optimization tasks. Third, moving beyond correctness toward improving the efficiency of formulations is crucial for real-world deployment, and our self-improving library learning approach offers a promising path toward that goal.

ETHICS STATEMENT

This research focuses on developing AlphaOPT, a self-improving experience library for optimization modeling. Our experiments are conducted entirely on publicly available optimization benchmarks and solver outputs. No personally identifiable information, human subject data, or sensitive demographic attributes are used. The solver feedback signals are purely algorithmic (e.g., feasibility, optimality gaps) and do not raise direct privacy concerns.

We do not identify other immediate ethical risks in the present work.

REPRODUCIBILITY STATEMENT

We are committed to ensuring the reproducibility of our results.

- **Datasets:** All experiments are based on standard, publicly available optimization benchmarks. References to each benchmark source are provided in the appendix.
- **Preprocessing:** Details on input normalization, task construction, and solver interfaces are described in the paper and appendix. Code for preprocessing and task generation will be released with the camera-ready version.
- **Algorithms and Models:** All implementation details, including training scripts, hyperparameter settings, and evaluation pipelines, will be released.

We will provide a public GitHub repository upon camera-ready submission that contains preprocessing scripts, the AlphaOPT library implementation, and end-to-end reproduction pipelines, enabling independent verification of all main results, figures, and tables reported in this paper.

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

APPENDIX

## A COMPARATIVE ANALYSIS OF ALPHAOPT AGAINST PRIOR EXPERIENCE-LEARNING METHODS

Recent approaches such as Reflexion (Shinn et al., 2023), STaR (Zelikman et al., 2022), ExpeL (Zhao et al., 2024), and AlphaEvolve (Novikov et al., 2025) demonstrate that large models can improve through experiential reuse, storing reflections, rationales, or code edits and applying them in new tasks. These methods have been effective in open-ended reasoning and programming, but they face limitations for optimization problems. First, their experiences are largely preserved as free-form text or edits without explicit applicability semantics, yet in optimization tasks, applying such experiences inappropriately can have detrimental effects. Second, their verification is limited to task outcomes such as checking rewards, final answers, or passing test cases, which does not guarantee that the underlying knowledge is structurally valid or transferable.

Our framework adapts experience learning to operations research (OR) with three key innovations: (1) solver-guided verifiability: correctness is judged at the program level. If a program achieves the optimal objective under the solver, it is highly likely to be valid and can serve as a reliable anchor for extracting insights, broadening the sources of experience collection. New and refined insights are explicitly re-tested on associated tasks, ensuring they are valid before integration; (2) structured knowledge for interpretability and auditability: each insight is represented with taxonomy, condition, explanation, and example, making its applicability explicit, reviewable, and even revisable in practice; (3) refinement of experience applicability for generalizability and preciseness: applicability conditions are refined using cross-task evidence, so insights neither over-generalize nor remain too narrow, improving safe transfer across problem families. See Table 3 for detailed comparisons.

## B DATASETS

We have collected the publicly available optimization problem datasets listed in the table, which include both natural language problem descriptions and their optimal solutions.

Because our library-based framework derives knowledge feedback from correct solutions, it is relatively sensitive to data noise. Accordingly, we train and evaluate on clean splits that exclude instances labeled as erroneous. Specifically, for NLP4LP, IndustryOR, and ComplexOR we use the cleaned versions provided by (Yang et al., 2025a); for NL4OPT, MAMO (EasyLP), MAMO (ComplexLP), and Optibench we use the cleaned releases from (Astorga et al., 2025), obtained from their (GitHub repository).

Table 3: Comparison of experience-learning methods. Prior works improve through experiential reuse but rely on free-form knowledge and outcome-level verification. Our framework introduces structured insights, solver-guided verification, and refined applicability, which are crucial for OR.

| Method | Structured knowledge | Explicit applicability | Verification | Applicability refinement | Application domain |
|---|---|---|---|---|---|
| Reflexion | ✗ | ✗ | Reward signal | ✗ | General agents |
| STaR | ✗ | ✗ | Answer correctness | ✗ | QA / reasoning |
| ExpeL | ✗ | (✓) minimal | Task success assumed | ✗ | General agents |
| AlphaEvolve | ✗ | (✓) implicit | Test harness (partial) | ✗ | Code synthesis / evolution |
| **AlphaOPT** | ✓ | ✓ | Solver optimality + insight verification | ✓ | OR formulation and solver code |

Table 4: The statistics of the optimization problem datasets

| Dataset | Size | Formulation Type(s) | Completion |
|---|---|---|---|
| NL4OPT (Ramamonjison et al., 2022) | 289 | LP | solution |
| NLP4LP (AhmadiTeshnizi et al., 2024) | 269 | LP, MILP, MINLP | solution, program |
| MAMO (complex LP and Easy LP) (Huang et al., 2024) | 863 | LP | solution |
| IndustryOR (Huang et al., 2025) | 100 | LP, IP, MILP, NLP, others | solution, program |
| Optibench (Yang et al., 2024) | 605 | LP, MILP, MINLP | solution |
| LogiOR (Yang et al., 2025a) | 80 | LP, IP, MIP, NLP | solution, program |

Abbreviations: LP – Linear Programming; IP - Integer Programming; NLP – Nonlinear Programming; MI – Mixed-Integer; others - Quadratic Programming, Dynamic&Stochastic Programming, etc.

## C SUCCESS AND FAILURE CASE STUDY

To better understand the effectiveness of insights in the experience library when solving new optimization problems, we conducted success and failure case studies on the test datasets. Success cases refer to instances where the agent successfully retrieved applicable insights that guided it to correctly solve previously failed problems. Failure cases refer to instances where the retrieved insights failed to help or even misled the agent, resulting in incorrect solutions.

### C.1 SUCCESS CASE ANALYSIS

In the testing datasets, most successful cases focus on correcting Variable Definition Errors, which identify that decision variables representing physically indivisible items should be integers. This highlights a common LLM failure point shared across training and test sets. Other common corrections include Objective Function Formulation Errors (such as confusing sum with makespan) and Constraint Formulation Errors (such as incorrect relational operators or unit inconsistency). Additionally, a few cases address more advanced or complex model structure errors.

### C.2 FAILURE CASE ANALYSIS

We summarize the failure cases into the following three representative types:

1) Overly narrow applicability: Rules induced from specific tasks may be over-generalized to different contexts (e.g., requiring "food variables must be integers" in nutrition or blending problems where continuous quantities are appropriate), which unnecessarily restricts the feasible region and eliminates valid solutions.

2) Mechanical enforcement of equalities/flow conservation: In unbalanced supply–demand settings or in the presence of surplus, rigidly enforcing == constraints or per-node "net inflow = demand" creates contradictions and renders the model infeasible.

3) Insight gaps due to data distribution: Limited coverage in training/retrieval samples (e.g., absence of multi-commodity vehicle routing cases with shared capacity constraints) can leave critical insights missing, leading to relaxed or mis-specified constraints.

The first two issues can be effectively mitigated in our framework through iterative recall and rewriting of tasks that expose misaligned insights, thereby refining their applicability boundaries. For the third issue, however, we call on the community to contribute broader and more diverse datasets to ensure wider applicability.

## D PROOF OF THE LIBRARY CONVERGENCE

Recall the optimization problem in the library training phase

$$F(\ell) \; = \; \mathbb{E}_{t \sim \mathcal{T}_{\text{train}}}\big[\, r(t \mid \ell)\,\big] \; - \; \lambda\,\Omega(\ell),$$

where $r(t, \ell)$ is a bounded reward function that implements the role of the original $\text{Success}(t \mid \ell)$ (i.e., it measures the matching quality between optimization problem $t$ and library $\ell$), and $\Omega(\ell)$ is a bounded complexity penalty.

According to the problem setting, the iterative refinement algorithm satisfies:

1. Monotone update: At iteration $k$, from $\ell_k$, the algorithm considers a set of admissible refinements $R(\ell_k) \subseteq \mathcal{L}$. Each accepted iteration consists of one of two types of operations:
   - *Merge step:* decreases $\Omega(\ell)$ while leaving $r(t, \ell)$ non-decrease for all relevant tasks;
   - *Exploration step:* improves $r(t, \ell)$ for some tasks without increasing $\Omega(\ell)$.

   Therefore every accepted refinement strictly increases $F(\ell)$; otherwise the algorithm keeps $\ell_{k+1} = \ell_k$.

2. Sufficient exploration: Any improving neighbor $\tilde{\ell} \in R(\ell_k)$ (i.e. one with strictly larger objective) will eventually be discovered and executed. Empirically, this is achieved through iterative prompt optimization with LLMs.

3. Boundedness: $r(t, \ell)$ and $\Omega(\ell)$ are bounded, hence $F(\ell)$ is bounded above and below.

The following theorem establishes that, under the assumption that the training and testing distributions are identical, the refinement procedure yields libraries that are locally optimal for the testing objective.

**Theorem 1.** *Assume $\mathcal{T}_{\mathrm{train}} = \mathcal{T}_{\mathrm{test}}$. If the library space $\mathcal{L}$ is finite, then the algorithm terminates in finitely many steps at a library $\ell^*$ which is a local maximizer for the testing objective. Moreover, the algorithm cannot terminate at a saddle point.*

*Proof.* Every accepted merge or exploration step strictly increases $F(\ell)$, and otherwise the library remains unchanged. Since $F$ is bounded above, the sequence $\{F(\ell_k)\}$ is monotone non-decreasing and bounded, hence convergent to some limit $F^*$. Furthermore, since $\mathcal{L}$ is finite, define

$$\delta = \min\left\{F(\tilde{\ell}) - F(\ell) : \tilde{\ell} \in R(\ell),\ F(\tilde{\ell}) > F(\ell)\right\}.$$

Finiteness guarantees $\delta > 0$, so only finitely many strict improvements are possible. The algorithm halts at some $\ell^*$. By sufficient exploration, no improving neighbor of $\ell^*$ exists. Therefore, $\ell^*$ is a local maximizer for both training and testing objectives. Saddle points are excluded.

Since the training and testing distributions coincide, the training objective equals the testing objective, so any local optimality statement directly applies to testing. □

Although the assumption of a finite library is reasonable, we also provide a proof for the case of an infinite library for completeness and rigor.

**Theorem 2** (Infinite compact library case). *Assume $\mathcal{T}_{\mathrm{train}} = \mathcal{T}_{\mathrm{test}}$. If the library space $\mathcal{L}$ is compact (closed and bounded) and $F$ is continuous, then the sequence $\{F(\ell_k)\}$ converges, and any subsequential limit point $\ell^\infty$ is a local maximizer for the testing objective. Saddle points are excluded for all such limit points.*

*Proof.* Each accepted step strictly increases $F(\ell)$, so $\{F(\ell_k)\}$ is monotone non-decreasing. Since $F$ is bounded above, $\{F(\ell_k)\}$ converges to some $F^*$. By compactness of $\mathcal{L}$, there exists a convergent subsequence $\ell_{k_j} \to \ell^\infty$. Continuity of $F$ ensures $F(\ell_{k_j}) \to F(\ell^\infty) = F^*$. Suppose $\ell^\infty$ had a neighbor $\tilde{\ell} \in R(\ell^\infty)$ with $F(\tilde{\ell}) > F(\ell^\infty)$. Then sufficient exploration would eventually yield $F(\ell_k) > F^*$, which is a contradiction. Therefore, $\ell^\infty$ is a local maximizer. Saddle points are excluded. Since the training and testing distributions coincide, the training objective equals the testing objective, so any local optimality statement directly applies to testing. □

Theorems 1 and 2 together guarantee that, when the training and testing distributions coincide, the refinement algorithm converges to locally optimal solutions for the testing phase.

## E  LIBRARY DISTRIBUTION ANALYSIS

### E.0.1  LIBRARY STRUCTURE AND COMPOSITION

**Domain Modeling.**  The distribution is relatively diffuse: Resource Allocation (34.78%) and Network Flow (27.54%) dominate, followed by Production Planning (15.94%). The remainder forms

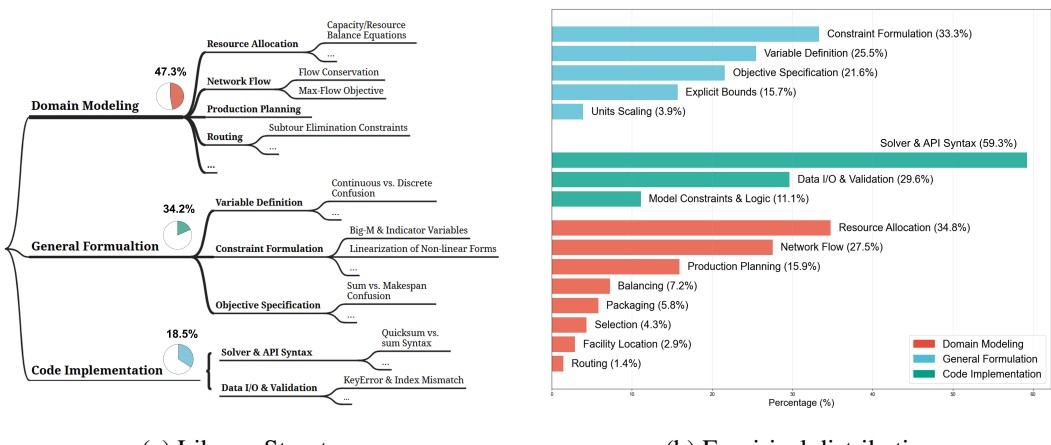

(a) Library Structure        (b) Empirical distribution

Figure 5: Library Content on Three Tracks

a long tail—Balancing (7.25%), Packaging (5.80%), Selection (4.35%), Facility Location (2.90%), and Routing (1.4%). Overall, Resource Allocation and Network Flow together account for over 60%, reflecting not only the composition of the training data but also the fact that these families harbor denser modeling pitfalls (e.g., flow-conservation/capacity coupling, which yield more failures and thus more recoverable insights.

**General Formulation.** The distribution is more concentrated: Constraint Formulation (33.33%), Variable Definition (25.49%), and Objective Specification (21.57%) sum to ∼80.39%, indicating that most errors arise when formulating constraints, defining variables, and specifying objectives. Explicit bounds (15.69%) and unit scaling (3.92%) are less frequent but persistent, pointing to recurring, codifiable issues for which reusable correction patterns exist.

**Code Implementation.** The distribution is markedly imbalanced, led by Solver & API Syntax (59.26%), followed by Data I/O & Validation (29.63%). This suggests that the primary pain points in generating and implementing optimization code lie in solver/API invocation and data ingestion/-validation.

### E.0.2 Insight Utility and Quality Analysis

The utility of insights lies in its ability to abstract and distill general modeling principles applicable to a wide range of optimization problems. For instance, the insight on Incorrect Relational Operators emphasizes the necessity of using the correct operators ($\geq$ or $\leq$) when translating natural language phrases like "at least" or "at most" into algebraic inequalities. This helps modelers fundamentally avoid errors and ensures the logical rigor of the model.

The quality of the insights is demonstrated by their in-depth analysis of specific, advanced modeling practices. For example, the insight into Knapsack Constraints highlights the importance of recognizing the hidden knapsack structure within a problem. This insight allows for the classification of a specific problem into a known optimization paradigm, enabling the more effective application of established solution strategies. Another example is the insight on Big-M Magnitude & Numerical Stability, which represents a high-quality practical guide. It delves into the algorithmic layer of solvers, revealing the critical impact of choosing an appropriate M value on solving efficiency and numerical stability. For instance, it suggests using a tighter M value like the maximum demand instead of an arbitrarily large number. This provides modelers with key guidance on how to construct mixed-integer programming models that are not only correct but also robust.

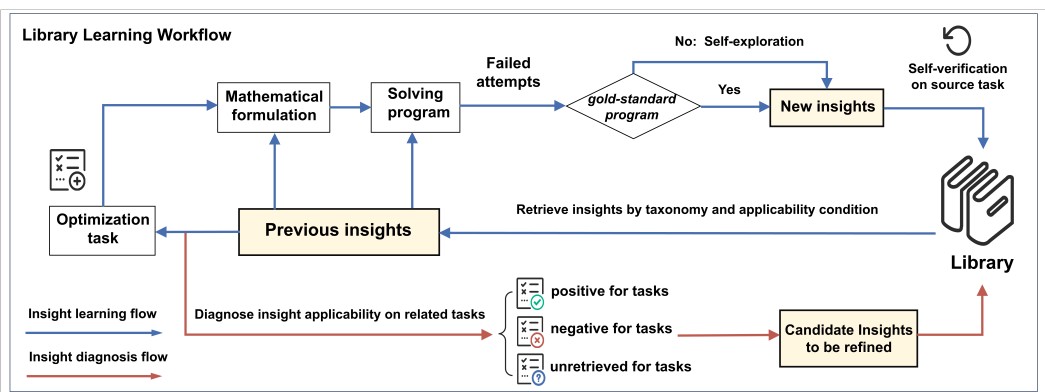

Figure 6: For each individual problem, **AlphaOPT** first proposes a formulation and checks its correctness. If the formulation is incorrect, it either analyzes and extracts insights from the gold-standard program (when available) or continues exploring solutions until it finds a correct answer, using the final generated formulation as a reference for insight extraction. The system then merges insights into the library in an online manner, performing self-verification before merging to ensure that the insights actually solve the source problems. For each insight, **AlphaOPT** also conducts diagnosis to identify related positive, negative, and unretrieved tasks, thereby preparing for refinements in later stages.

Table 5: The ablation results of AlphaOPT without: performance on benchmarks.

| Dataset | Logior | Optibench | MAMO-Easy |
|---|---|---|---|
| AlphaOPT (full) | 51.08% | 91.81% | 95.59% |
| w/o self-debug | 35.87% | 89.26% | 95.06% |
| w/o taxonomy | 46.74% | 90.72% | 94.49% |
| w/o insight example | 45.65% | 91.06% | 95.06% |

# F  IMPLEMENT DETAILS AND ADDITIONAL RESULTS

## F.1  LIBRARY LEARNING WORKFLOW

## F.2  LIBRARY REFINEMENT FRAMEWORK

## F.3  ABLATION STUDY

We assess the effectiveness of library insight retrieval/application and self-debug:

- **w/o self-debug:** Remove the model's self-debugging section.

- **w/o taxonomy:** Remove the library taxonomy and directly match insights by checking all conditions.

- **w/o insight example:** Use only the explanation as input, excluding exemplar cases

Table 5 shows that the full AlphaOPT achieves the best scores on all datasets. Removing self-debug yields the largest drop on Logior (15.21%), indicating that iterative self-correction plays an important role in the system. Dropping the library taxonomy reduces accuracy by 4.34% on Logior and by 1.09% on Optibench, suggesting that structured matching meaningfully improves retrieval. Excluding insight examples similarly lowers performance, showing that concrete, worked snippets aid application beyond textual explanations alone. Results on MAMO-Easy are nearly unchanged across ablations ($\leq 0.53\%$), implying a ceiling effect on easier instances.

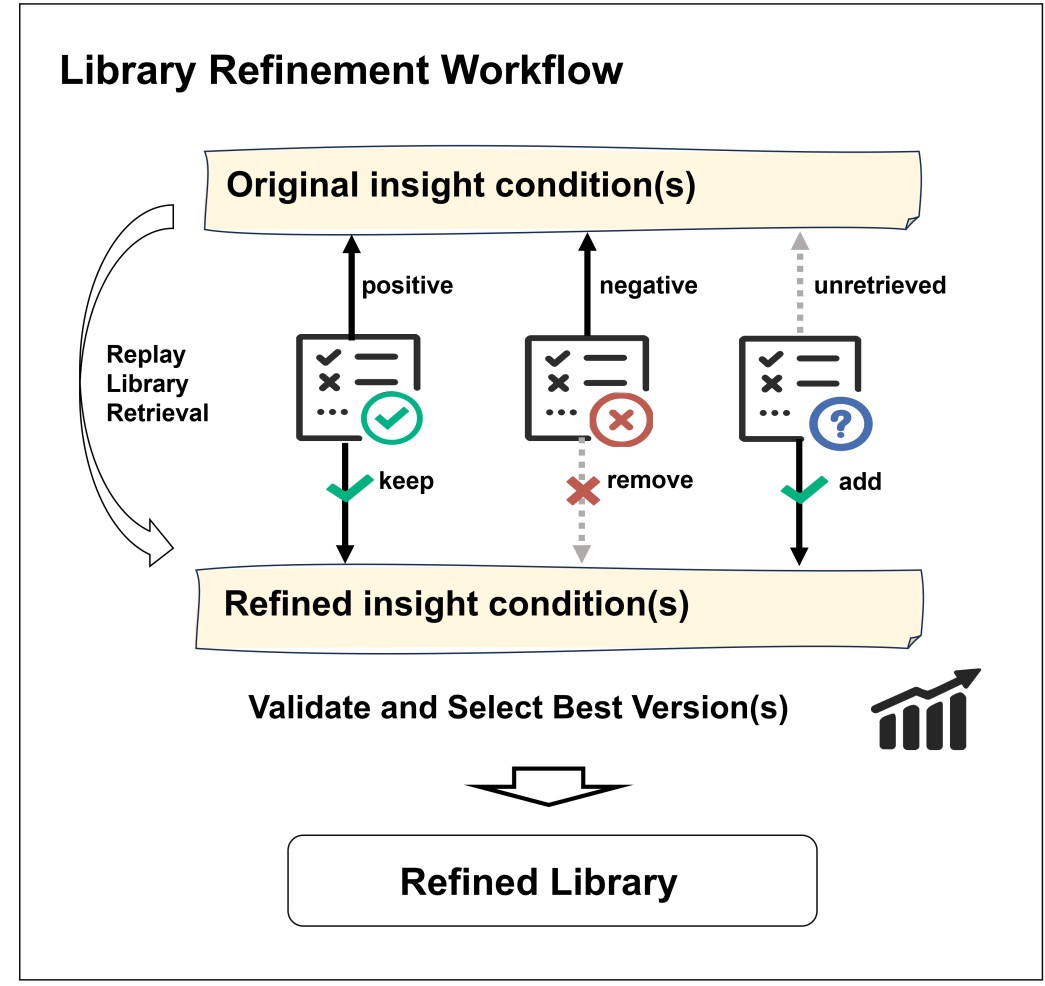

Figure 7: AlphaOPT introduces a library-refinement mechanism that revises the applicability conditions of insights diagnosed as misaligned on prior tasks that retrieved them.

### F.4 PROMPTS FOR LLM MODULES

**Apply self-explore on finding gold-standard program**

```
You are an expert in Industrial Engineering and Operations Research
    teaching graduate students to avoid modeling-and-coding mistakes
    in solving optimization problems.

You are given:
1. A problem description for the optimization task
2. A mathematical model proposed by your colleague which failed to
    yield an optimal solution when solved with the Gurobi optimizer (
    hereafter referred to as *the failed mathematical model*)
3. The gold-standard program, which embodies the correct mathematical
    formulation of the optimization task

### Problem description
{problem_description}
```

```
### The failed mathematical model
(Note: the model is written in LaTeX and presented in a plain-text
    code block (```))
{failed_formulation}

### The gold-standard program
{correct_program}

### Your task
Step 1: Compare the failed mathematical model with the correct
    mathematical model embodied in the gold-standard program to
    identify issues that prevent optimality. Note that variable names
    in the proposed model may differ from those in the gold-standard
    program. Please align them carefully based on the problem
    specification.

Step 2: Using the insight taxonomy dictionaries provided below,
    extract one or more new insights, which should be a distinct and
    concise lesson derived from a specific issue identified in the
    failed mathematical model relative to the gold-standard program.

Each new insight must contain exactly four fields:

1) **taxonomy**  choose **exactly one** of the two aspects:
    - **Domain Modeling**: Level-1 = Problem Domain (e.g., "Network
    Flow"); Level-2 = Specific Technique (e.g., "Flow Conservation").
    - **General Formulation**: Level-1 = Formulation Component (e.g.,
    "Variable Definition"); Level-2 = Specific Aspect/Pitfall (e.g., "
    Continuous vs. Discrete Confusion").

    Taxonomy rule (nested-dict): `{{ Level-1 : {{ Level-2 : null | {{
    "definition": "...", "condition": "..." }} }} }}`
    - The taxonomy MUST always be expressed as a **three-level nested
    dict**: Level-1  Level-2  (null or a dict with "definition"/"
    condition").
    - Pick **exactly one** Level-1 (existing key).
    - Pick **one or more** Level-2 under that Level-1 (existing key or
     keys).
    - For an existing Level-2, set its value to null.
    - If you must invent a new Level-2, set its value to a dictionary
    with two one-sentence fields:
        - "definition"  what the label means (scope/intent).
        - "condition"  when to apply the label (a general trigger
    grounded in the problem description or in the defining features of
     the problem domain).
    - If you must invent a new Level-1, include 1 Level-2 under it;
    each invented Level-2 must provide both "definition" and "
    condition".

2) **condition**  Write it as a trigger explicitly grounded in the
    problem description or in the defining features of the problem
    domain. First state the general situation, then use this problem
    as an example. **Use the pattern below**, and keep it strictly non
    -prescriptive: do not give any solution, advice or decision:
"This insight applies when ... For example, when the problem statement
     mentioned ...".

3) **explanation**  A brief and self-contained description that
    specifies, under this condition, what the best practice is, what
    the common mistake is and its cause. First, use this problem as an
```

```
        example to illustrate; Then, appropriately generalize the correct
        modeling strategy it reflects, if applicable.
  **Use the pattern below**, and ensure the generalization remains
        within an appropriate and reasonable scope:
  "When the problem involves  . The best practice is  . A common mistake
        is  , which happens because  . More generally, this reflects that
         ."

  4) **example**  A brief, self-contained demonstration showing wrong vs
        . correct version (principle, formulation, or code snippet).
        Clearly mark them as '# Wrong' and '# Correct'.

  ### Taxonomy Dictionaries
  **Domain Modeling**
  {domain_taxo}

  **General Formulation**
  {formulation_taxo}

  ### STRICT OUTPUT FORMAT
  Return a single JSON **array** of insight objects. No text before/
        after. Example with two insights (but not must be two):

  [
      {{
          "taxonomy": {{
              "Domain Modeling": {{
                  "Network Flow": {{
                      "<New Label If Necessary>": {{ "definition": "<one
      sentence>", "condition": "<one sentence>" }}
                  }}
              }}
          }},
          "condition": "<text>",
          "explanation": "<text>",
          "example": "<text>"
      }},

      {{
          "taxonomy": {{
              "General Formulation": {{
                  "Variable Definition": {{
                      "Continuous vs. Discrete Confusion": null
                  }}
              }}
          }},
          "condition": "<text>",
          "explanation": "<text>",
          "example": "<text>"
      }}
  ]

  **Guidelines**:
  - Output as many **distinct, non-overlapping** insights as needed.
  - Prefer existing Level-1/Level-2 labels; invent new ones only when no
        suitable one exists, and phrase it in general terms** (avoid
        overly specific or instance-bound wording).
  - **Be precise in stage selection**use **Domain Modeling** for domain-
        specific techniques that arise only within specific problem
        families (e.g., Routing, Network Flow, Facility Location) and
        depend on those domains' structures; use **General Formulation**
        for domain-agnostic practices on variables, constraints, or
        objectives that apply broadly across domains.
```

Now take a deep breath and think step by step.

## Generate structured insights

You are an expert in Industrial Engineering and Operations Research
    teaching graduate students to avoid modeling-and-coding mistakes
    in solving optimization problems.

You are given:
1. A problem description for the optimization task
2. A mathematical model proposed by your colleague which failed to
    yield an optimal solution when solved with the Gurobi optimizer (
    hereafter referred to as *the failed mathematical model*)
3. The gold-standard program, which embodies the correct mathematical
    formulation of the optimization task

### Problem description
{problem_description}

### The failed mathematical model
(Note: the model is written in LaTeX and presented in a plain-text
    code block (```))
{failed_formulation}

### The gold-standard program
{correct_program}

### Your task
Step 1: Compare the failed mathematical model with the correct
    mathematical model embodied in the gold-standard program to
    identify issues that prevent optimality. Note that variable names
    in the proposed model may differ from those in the gold-standard
    program. Please align them carefully based on the problem
    specification.

Step 2: Using the insight taxonomy dictionaries provided below,
    extract one or more new insights, which should be a distinct and
    concise lesson derived from a specific issue identified in the
    failed mathematical model relative to the gold-standard program.

Each new insight must contain exactly four fields:

1) **taxonomy**  choose **exactly one** of the two aspects:
    - **Domain Modeling**: Level-1 = Problem Domain (e.g., "Network
    Flow"); Level-2 = Specific Technique (e.g., "Flow Conservation").
    - **General Formulation**: Level-1 = Formulation Component (e.g.,
    "Variable Definition"); Level-2 = Specific Aspect/Pitfall (e.g., "
    Continuous vs. Discrete Confusion").

    Taxonomy rule (nested-dict): `{{ Level-1 : {{ Level-2 : null | {{
    "definition": "...", "condition": "..." }} }} }}`
    - The taxonomy MUST always be expressed as a **three-level nested
    dict**: Level-1  Level-2  (null or a dict with "definition"/"
    condition").
    - Pick **exactly one** Level-1 (existing key).

```
    - Pick **one or more** Level-2 under that Level-1 (existing key or
     keys).
    - For an existing Level-2, set its value to null.
    - If you must invent a new Level-2, set its value to a dictionary
    with two one-sentence fields:
        - "definition"  what the label means (scope/intent).
        - "condition"  when to apply the label (a general trigger
    grounded in the problem description or in the defining features of
     the problem domain).
    - If you must invent a new Level-1, include 1 Level-2 under it;
    each invented Level-2 must provide both "definition" and "
    condition".

2) **condition**  Write it as a trigger explicitly grounded in the
    problem description or in the defining features of the problem
    domain. First state the general situation, then use this problem
    as an example. **Use the pattern below**, and keep it strictly non
    -prescriptive: do not give any solution, advice or decision:
"This insight applies when ... For example, when the problem statement
    mentioned ...".

3) **explanation**  A brief and self-contained description that
    specifies, under this condition, what the best practice is, what
    the common mistake is and its cause. First, use this problem as an
     example to illustrate; Then, appropriately generalize the correct
     modeling strategy it reflects, if applicable.
**Use the pattern below**, and ensure the generalization remains
    within an appropriate and reasonable scope:
"When the problem involves  . The best practice is  . A common mistake
     is  , which happens because  . More generally, this reflects that
      ."

4) **example**  A brief, self-contained demonstration showing wrong vs
    . correct version (principle, formulation, or code snippet).
    Clearly mark them as '# Wrong' and '# Correct'.

### Taxonomy Dictionaries
**Domain Modeling**
{domain_taxo}

**General Formulation**
{formulation_taxo}

### STRICT OUTPUT FORMAT
Return a single JSON **array** of insight objects. No text before/
    after. Example with two insights (but not must be two):

[
    {{
        "taxonomy": {{
            "Domain Modeling": {{
                "Network Flow": {{
                    "<New Label If Necessary>": {{ "definition": "<one
    sentence>", "condition": "<one sentence>" }}
                }}
            }}
        }},
        "condition": "<text>",
        "explanation": "<text>",
        "example": "<text>"
    }},

    {{
```

```
          "taxonomy": {{
              "General Formulation": {{
                  "Variable Definition": {{
                      "Continuous vs. Discrete Confusion": null
                  }}
              }}
          }},
          "condition": "<text>",
          "explanation": "<text>",
          "example": "<text>"
      }}
  ]

  **Guidelines**:
  - Output as many **distinct, non-overlapping** insights as needed.
  - Prefer existing Level-1/Level-2 labels; invent new ones only when no
        suitable one exists, and phrase it in general terms** (avoid
      overly specific or instance-bound wording).
  - **Be precise in stage selection**use **Domain Modeling** for domain-
      specific techniques that arise only within specific problem
      families (e.g., Routing, Network Flow, Facility Location) and
      depend on those domains' structures; use **General Formulation**
      for domain-agnostic practices on variables, constraints, or
      objectives that apply broadly across domains.

  Now take a deep breath and think step by step.
```

**Diagnose Issues for failed program**

```
You are an expert in Industrial Engineering and Operations Research.

You are given:
1. A problem description for the optimization task
2. A mathematical model proposed by your colleague which failed to
    yield an optimal solution when solved with the Gurobi optimizer (
    hereafter referred to as *the failed mathematical model*)
3. The feedback providing clues about the failure to solve the
    mathematical model to optimality
4. The gold-standard program, which embodies the correct mathematical
    formulation of the optimization task

### Problem description
{problem_description}

### The failed mathematical model
Note: the model is written in LaTeX and presented in a plain-text code
    block (''')
{failed_formulation}

### The feedback
{feedback}

### The gold-standard program
{correct_program}

### Your task
```

```
Step 1: Compare the failed mathematical model with the correct one
    embodied in the gold-standard program, and identify all
    formulation issues that prevent optimality. Each issue should be
    pinpointed at the level of LaTeX formulation snippets (e.g.,
    variables, constraints, and the objective function), and should
    correspond to a single, well-defined correction point. Note that
    variable names in the proposed model may differ from those in the
    gold-standard program, so please align them carefully based on the
     problem specification.

Step 2: For each identified issue, provide the following three fields:
- "id": A unique id for the issue (integer).
- "issue": A concise description of the issue.
- "evidence": The evidence showing where the issue occurs, including
    the excerpt from the failed mathematical model (mark as #wrong)
    and the corresponding excerpt from the gold-standard program (mark
     as #correct).

Step 3: Minimize overlap by reporting **independent, root-cause issues
    **. If multiple defects share the same fix point or strategy,
    merge them into a single composite issue. If several issues are
    upstream/downstream symptoms of the same root cause (i.e., they
    would be fixed by the same correction), consolidate them into one
    composite issue.

### STRICT OUTPUT FORMAT
**Return only a JSON array** of your answer. Each array element must
    be an object with keys '"id"', '"issue"' and '"evidence"'.

Example:

```json
[
    {{"id": 1,"issue": "...", "evidence": "..."}},
  {{"id": 2,"issue": "...", "evidence": "..."}}
]
```

**Guidelines:**
- Make sure to identify **distinct and independent issues** (e.g.,
    missing constraints, wrong variable bounds, or incorrect objective
     formulation).
- Do NOT include issues that do not directly affect the model's
    ability to reach optimality.
- Only output the JSON array. DO NOT include any explanations,
    markdown, or extra text before or after the JSON array.

Now take a deep breath and think step by step.
```

**Diagnose Issues**

```
You are an expert in Industrial Engineering and Operations Research.

You are given:
1. A problem description for the optimization task
2. A mathematical model proposed by your colleague which failed to
    yield an optimal solution when solved with the Gurobi optimizer (
    hereafter referred to as *the failed mathematical model*)
```

```
3. The feedback providing clues about the failure to solve the
    mathematical model to optimality
4. The gold-standard program, which embodies the correct mathematical
    formulation of the optimization task

### Problem description
{problem_description}

### The failed mathematical model
Note: the model is written in LaTeX and presented in a plain-text code
    block (''')
{failed_formulation}

### The feedback
{feedback}

### The gold-standard program
{correct_program}

### Your task

Step 1: Compare the failed mathematical model with the correct one
    embodied in the gold-standard program, and identify all
    formulation issues that prevent optimality. Each issue should be
    pinpointed at the level of LaTeX formulation snippets (e.g.,
    variables, constraints, and the objective function), and should
    correspond to a single, well-defined correction point. Note that
    variable names in the proposed model may differ from those in the
    gold-standard program, so please align them carefully based on the
     problem specification.

Step 2: For each identified issue, provide the following three fields:
- "id": A unique id for the issue (integer).
- "issue": A concise description of the issue.
- "evidence": The evidence showing where the issue occurs, including
    the excerpt from the failed mathematical model (mark as #wrong)
    and the corresponding excerpt from the gold-standard program (mark
     as #correct).

Step 3: Minimize overlap by reporting **independent, root-cause issues
    **. If multiple defects share the same fix point or strategy,
    merge them into a single composite issue. If several issues are
    upstream/downstream symptoms of the same root cause (i.e., they
    would be fixed by the same correction), consolidate them into one
    composite issue.

### STRICT OUTPUT FORMAT
**Return only a JSON array** of your answer. Each array element must
    be an object with keys '"id"', '"issue"' and '"evidence"'.

Example:

```json
[
    {{"id": 1,"issue": "...", "evidence": "..."}},
  {{"id": 2,"issue": "...", "evidence": "..."}}
]
```

```
```

**Guidelines:**
- Make sure to identify **distinct and independent issues** (e.g.,
    missing constraints, wrong variable bounds, or incorrect objective
     formulation).
- Do NOT include issues that do not directly affect the model's
    ability to reach optimality.
- Only output the JSON array. DO NOT include any explanations,
    markdown, or extra text before or after the JSON array.

Now take a deep breath and think step by step.
```

---

### Diagnose Issues

```
You are an expert in Industrial Engineering and Operations Research.

You are given:
1. A problem description for the optimization task
2. A mathematical model proposed by your colleague which failed to
    yield an optimal solution when solved with the Gurobi optimizer (
    hereafter referred to as *the failed mathematical model*)
3. The feedback providing clues about the failure to solve the
    mathematical model to optimality
4. The gold-standard program, which embodies the correct mathematical
    formulation of the optimization task

### Problem description
{problem_description}

### The failed mathematical model
Note: the model is written in LaTeX and presented in a plain-text code
    block (```)
{failed_formulation}

### The feedback
{feedback}

### The gold-standard program
{correct_program}

### Your task

Step 1: Compare the failed mathematical model with the correct one
    embodied in the gold-standard program, and identify all
    formulation issues that prevent optimality. Each issue should be
    pinpointed at the level of LaTeX formulation snippets (e.g.,
    variables, constraints, and the objective function), and should
    correspond to a single, well-defined correction point. Note that
    variable names in the proposed model may differ from those in the
    gold-standard program, so please align them carefully based on the
     problem specification.

Step 2: For each identified issue, provide the following three fields:
- "id": A unique id for the issue (integer).
- "issue": A concise description of the issue.
```

```
 – "evidence": The evidence showing where the issue occurs, including
     the excerpt from the failed mathematical model (mark as #wrong)
     and the corresponding excerpt from the gold-standard program (mark
      as #correct).

Step 3: Minimize overlap by reporting **independent, root-cause issues
    **. If multiple defects share the same fix point or strategy,
    merge them into a single composite issue. If several issues are
    upstream/downstream symptoms of the same root cause (i.e., they
    would be fixed by the same correction), consolidate them into one
    composite issue.

### STRICT OUTPUT FORMAT
**Return only a JSON array** of your answer. Each array element must
    be an object with keys `"id"`, `"issue"` and `"evidence"`.

Example:

```json
[
    {{"id": 1,"issue": "...", "evidence": "..."}},
  {{"id": 2,"issue": "...", "evidence": "..."}}
]
```

**Guidelines:**
- Make sure to identify **distinct and independent issues** (e.g.,
    missing constraints, wrong variable bounds, or incorrect objective
     formulation).
- Do NOT include issues that do not directly affect the model's
    ability to reach optimality.
- Only output the JSON array. DO NOT include any explanations,
    markdown, or extra text before or after the JSON array.

Now take a deep breath and think step by step.
```

# G    USE OF LARGE LANGUAGE MODELS

We used Large Language Models for two main purposes: first, to debug our code during the development of the agent system, and second, to check for grammar mistakes in our writing.

