# OpenReview forum: "AlphaOPT: Formulating Optimization Programs with Self-Improving LLM Experience Library"
_ICLR.cc/2026/Conference — ICLR 2026 Conference Withdrawn Submission_

### Official Review · Reviewer_MDu6 · 2025-10-30

**Soundness:** 3
**Presentation:** 3
**Contribution:** 3
**Rating:** 4
**Confidence:** 4

**Summary:**

AlphaOPT presents a conceptually interesting and well-motivated approach that reframes optimization modeling for LLMs as an evolving knowledge-base problem rather than parameter learning. The paper is technically sound, and the empirical evidence supports the main claims. However, the evolutionary refinement process remains somewhat heuristic, and the solver feedback pipeline could be more formally characterized in terms of sample efficiency and stability. Please refer to an example of previous work (Voyager: An Open-Ended Embodied Agent with Large Language Models) that shares the same spirit with AlphaOPT.

**Strengths:**

Problem significance and framing: The work addresses a major bottleneck in applying LLMs to operations research and mathematical optimization—translating informal language into executable solver programs. The authors clearly articulate why both prompt-based (fragile templates) and fine-tuned (data-hungry, low-transfer) systems fail to generalize, motivating a self-improving paradigm through experience collection.

Empirical results and generalization: Table 2 and Fig. 4 show strong out-of-distribution gains (e.g., +7.7% over LLMOPT on OptiBench). AlphaOPT also demonstrates answer-only learning (Sec. 4.3) and monotonic performance growth with more data (Table 1). These results substantiate the continual-learning claim and highlight test-time adaptation.

**Weaknesses:**

Novelty concern: This is a very important concern. AlphaOPT’s framing, i.e., “self-improving experience library” for optimization modeling, sounds conceptually new, but the underlying mechanics draw heavily from existing reflection and retrieval-based self-improvement paradigms, such as Reflexion (Shinn et al., 2023), AlphaEvolve / ReEvo (Novikov et al., 2025; Ye et al., 2024), Voyager (Wang et al, 2023). OPTIMIZATION PERSPECTIVE (sec 3.2) is also similar to a very recent work that uses a maximum set covering to prune redundant rules (WALL-E, Zhou et al 2024).

So the core novelty is more domain-adaptational than algorithmic: they transplant self-reflection + retrieval ideas into operations research formulation, with solver verification as a robust correctness oracle. That’s valuable engineering work but arguably not a foundational methodological leap.

**Questions:**

I would like to know whether the growth of library easily hits the limitation of LLM's context length. And I think memory/context compression (e.g., KV merging) can be more efficient, accurate, and promising than RAG + library reduction.

---

### Official Review · Reviewer_igsk · 2025-11-01

**Soundness:** 2
**Presentation:** 3
**Contribution:** 3
**Rating:** 4
**Confidence:** 4

**Summary:**

AlphaOPT is a training-free method that proposes a two-phase, self-improving experience library for operations-research modeling: Library Learning extracts solver-verified insights with a schema of taxonomy, condition, explanation, example; Library Evolution diagnoses mis-retrievals and refines applicability conditions over iterations. The method reports answer-only (no gold program) gains and OOD generalization; the gains continue to grow as data and the library grow.

**Strengths:**

1) Good motivation. Learning from failed samples generated by the model during scaled inference, in a continuous way, is a good strategy to enhance accuracy.


2) Answers-only learning works and transfers OOD. The method operates well without gold programs (which are hard to acquire for new domains); learning solely from answers (self-explore) yields accuracy comparable to full supervision.


3) Continual macro gains with compact growth. From 100 to 300 training items, performance rises while the library remains modest in size.

**Weaknesses:**

1) Runtime/latency and token accounting are absent. The paper does not report retrieval or inference latency and token/cost budget, making it hard to assess practicality and to compare cost with alternative agentic/test-time scaling methods.



2) Reliance on solver feedback restricts applicability. Learning and verification hinge on a solver producing/validating optimal values; many real-world reasoning tasks lack such verifiers, limiting external validity beyond domains with executable solvers.
3) Multi-module complexity. The pipeline is a multi-stage, multi-module agentic system with no analysis of how much each stage contributes to end-to-end cost/latency.

4) Retrieval quality not quantified. Retrieval is described, but there are no accuracy metrics (e.g. precision/recall) reported.

**Questions:**

How does this method compare to reinforcement learning with verifiable feedback (RLVF) approaches or large reasoning models?

See weaknesses for other analysis requested.

---

### Official Review · Reviewer_qYdJ · 2025-11-01

**Soundness:** 2
**Presentation:** 1
**Contribution:** 2
**Rating:** 2
**Confidence:** 3

**Summary:**

The paper proposes to tackle optimization problems with LLMs. The method proposes to build an evolving experience library for the LLM to learn on, so that the performance increases over experience.

**Strengths:**

1. Works on the relatively less-explored optimization problems with LLMs.
2. Evidence showing that building an experience library helps the accuracy over time.

**Weaknesses:**

1. The math in 3.2 lacks information and proof in Appendix D contains too strong assumptions that makes the Theorem 1 meaninglessly trivial. The assumption 2 is far from being realistic, and cannot be guaranteed in the infinite case in Theorem 2.
1. Any evaluation or explanation on why applying LLMs to optimization problems brings benefits compared to using readily available tools? Is there a need for automating this? Otherwise, in what situations would people want to use LLMs? This question is important for justifying the effort devoted in this paper in the beginning.
1. Why are the data in LogiOR and OptiBench "out-of-distribution"? No sample overlap does not directly mean the distributions are different, as they could well be independent samples from the same distribution.

**Questions:**

1. Is there any constraint that limits this approach from being applied to other coding domains? The proposed framework looks general. If it is indeed generally applicable (with only minor adaptions, such as replacing the toxonomy + conditions to task descriptions + inputs in coding tasks), why is this technique applied to a rather specific domain first? If it is not generally applicable, what could be the reasons? Appendix A seems to touch on this problem but is very vague to clearly understand.
1. For fairer comparison with fine-tuning based baselines, can you compare the training resources with them? Excuse me but I didn't find information about what backbone model is used for your method. Can you point me to it? This is also related to a fair comparison.
1. Any sign that the model creats new tools, or only using existing ones? Applying LLMs to tasks that existing solutions can solve does not bring much benefit to the domain, but getting new tools would help more.

---

> ### Author Response · Authors · 2025-12-03
> **Poor Review Quality With Little Knowledge in the Relevant Topic**
>
> We thank the reviewer for spending the time and effort to review our paper. However, we are concerned about the quality of the review, as it indicates that the reviewer possesses very limited familiarity with this research area.
>
> Regarding the second weakness, the entire paper is motivated by the potential of large language models to assist and automate optimization-formulation workflows, which are critical components of real-world decision systems across logistics, retail, energy, manufacturing, and many other industries. We clearly articulate this motivation in both the abstract and the introduction, and we include an extensive literature review to contextualize our contribution. These discussions appear to have been overlooked by the reviewer.
>
> Regarding the first weakness, we explicitly state in the paper that “Given the inherent ambiguity of natural language and the stochasticity in LLM outputs, we present this perspective not as a strict theorem but as a principled justification for the acquisition–refinement design and the redundancy-reduction operations.” Our intention is to provide a conceptual framework that clarifies the design principles, limitations, and necessary capabilities of LLMs for building an effective experience library. The reviewer’s comment suggests a misunderstanding of this clearly stated message.
>
> Regarding the third weakness, LogiOR and OptiBench are distinct datasets proposed by different works, constructed from different data sources and data-generation processes. We encourage the reviewer to consult the relevant literature before attributing this as a weakness.
>
> Overall, we appreciate the opportunity to clarify these points. We hope the AC and future readers will consider that the reviewer’s comments stem from misinterpretations and lack of relevant knowledge rather than substantive issues with the technical content or contributions of our work. We respectfully maintain that our paper presents a well-motivated, clearly justified, and practically relevant contribution to the emerging field of LLM-based optimization formulation.

---

### Official Review · Reviewer_6Bxv · 2025-11-01

**Soundness:** 2
**Presentation:** 3
**Contribution:** 3
**Rating:** 4
**Confidence:** 3

**Summary:**

This paper introduces AlphaOPT, a self-improving experience library framework designed for optimization modeling. The core contribution is a two-phase "Library Learning" and "Library Evolution" cycle that iteratively builds and refines a structured, solver-verified library of modeling insights. The primary contributions are: (1) A novel framework that mitigates the high cost of model retraining by updating an external, explicit knowledge library rather than model weights; (2) The practical capability to learn from "answer-only" supervision through solver-guided self-exploration, which is far more scalable than relying on scarce gold-standard programs ; and (3) Demonstrating strong out-of-distribution (OOD) generalization, which validates the transferability and effectiveness of the learned insights.

**Strengths:**

Originality and Quality: While the high-level 'learn-from-experience' pipeline is conceptually familiar, the paper's originality stems from its rigorous, domain-specific adaptation to optimization modeling. The design of the structured 4-tuple insight and Library Evolution is well-suited for this domain.

Significance: The ability to learn effectively from 'answer-only' supervision is highly significant. This directly addresses a critical bottleneck in the field, as gold-standard formulation programs are far scarcer and more expensive to create than simple optimal answers. This feature substantially lowers the barrier to adopting and scaling such systems in real-world OR applications.

Clarity: The paper is well-written, clear, and the methodology is easy to follow. The two-stage framework is presented logically, and the figures (e.g., Figure 1 and 2) effectively illustrate the core mechanism of insight retrieval and refinement.

**Weaknesses:**

Unfair Experimental Comparison (Base Model Disparity): The primary weakness lies in the experimental comparisons in Figure 4 and Table 2. The paper positions AlphaOPT as a framework, but its performance is inherently tied to the underlying base LLM. Figure 1's reference to GPT-4o suggests AlphaOPT utilizes a state-of-the-art proprietary model. This model is orders of magnitude larger and more capable than the baselines used for comparison (i.e., LLaMa3-8B for ORLM and Qwen2.5-14B for LLMOPT). This vast difference in base model capacity is a major confounding variable. The observed performance gap in OOD generalization may be largely, or even entirely, attributable to the base model's superior reasoning capability rather than the methodological advantages of AlphaOPT over fine-tuning. The strong performance of other prompt-based baselines (which likely also use a strong base model) reinforces this concern.

Missing Relevant Baselines: The evaluation lacks comparisons against state-of-the-art methods of the same category. The paper contrasts AlphaOPT with prompt-only and fine-tuning methods but omits other experience-library or self-improvement frameworks. A significant body of work exists in related domains (e.g., code generation, agentic workflows, SWE-bench) that also uses self-reflection and library learning. A more compelling evaluation would involve adapting a SOTA method from that domain to optimization modeling to demonstrate that AlphaOPT's domain-specific design is indeed superior.


Lack of Cost-Benefit Analysis: The paper claims AlphaOPT avoids "costly retraining". While this is true for training costs, the framework likely incurs significant inference costs. The solver-guided self-exploration , diagnostic , and evolver phases appear to require numerous, complex LLM calls. A discussion of the inference budget (e.g., total tokens, API costs, or wall-clock time) required to build the library for 300 items is missing. This omission makes a comprehensive cost-benefit analysis against the one-time cost of fine-tuning difficult.

**Questions:**

What specific base LLM (and version, e.g., GPT-4o, GPT-4-Turbo) was used for all LLM modules (insight extraction, diagnosis, evolution, and final generation) within the AlphaOPT framework for the main experiments in Table 2 and Figure 4?

To provide a fair comparison and isolate the framework's contribution, could the authors provide an ablation study showing AlphaOPT's performance when its base model is restricted to the same models as the baselines (i.e., LLaMa3-8B or Qwen2.5-14B)? This would demonstrate how much of the performance gain is attributable to the AlphaOPT framework itself versus the base model's capacity.

Could the authors elaborate on the exclusion of baselines from the "experience library" or "self-improvement" category (e.g., methods from code repair or agent domains like SWE-bench)? How does AlphaOPT's domain-specific design (e.g., the 4-tuple insight and solver-verification) fundamentally differ from or improve upon applying a more general-purpose self-improvement framework to this task?

---

### Author Response · Authors · 2025-12-03

We sincerely thank reviewers 6Bxv, qYdJ, and MDu6 for their comments. Most of the comments are grounded in constructive observations, and we will address and incorporate them in our future submission. We believe these comments will significantly help us improve the clarity and quality of our work. However, the comments from reviewer qYdJ indicate a clear lack of knowledge in this field, and we are concerned about the quality and fairness of that particular review. Given this situation, we decide to withdraw the paper. We again thank the reviewers who provided thoughtful and helpful feedback.

---

### Note · Authors · 2025-12-30

I have read and agree with the venue's withdrawal policy on behalf of myself and my co-authors.